

# A comparison of rumen microbial profiles in dairy cows as retrieved by 454 Roche and Ion Torrent (PGM) sequencing platforms

Nagaraju Indugu[1], Kyle Bittinger[2], Sanjay Kumar[1], Bonnie Vecchiarelli[1] and Dipti Pitta[1]

[1] Department of Clinical Studies, University of Pennsylvania, Kennett Square, PA, United States of America
[2] Department of Microbiology and Department of Biostatistics and Epidemiology, University of Pennsylvania, Philadelphia, PA, United States of America

Corresponding author
Dipti Pitta, dpitta@vet.upenn.edu

## ABSTRACT

Next generation sequencing (NGS) technology is a widely accepted tool used by microbial ecologists to explore complex microbial communities in different ecosystems. As new NGS platforms continue to become available, it becomes imperative to compare data obtained from different platforms and analyze their effect on microbial community structure. In the present study, we compared sequencing data from both the 454 and Ion Torrent (PGM) platforms on the same DNA samples obtained from the rumen of dairy cows during their transition period. Despite the substantial difference in the number of reads, error rate and length of reads among both platforms, we identified similar community composition between the two data sets. Procrustes analysis revealed similar correlations ($M^2 = 0.319; P = 0.001$) in the microbial community composition between the two platforms. Both platforms revealed the abundance of the same bacterial phyla which were Bacteroidetes and Firmicutes; however, PGM recovered an additional four phyla. Comparisons made at the genus level by each platforms revealed differences in only a few genera such as *Prevotella*, *Ruminococcus*, *Succiniclasticum* and *Treponema* ($p < 0.05$; chi square test). Collectively, we conclude that the output generated from PGM and 454 yielded concurrent results, provided stringent bioinformatics pipelines are employed.

## INTRODUCTION

Microbes are integral components of a diverse group of ecosystems and have co-evolved with their host/habitat in a mutually symbiotic relationship (*Hobson & Stewart, 1997*). The foregut (rumen) of ruminants is comprised of a complex microbial genetic web (rumen microbiome) that plays a pivotal role in the host nutrition and ultimate wellbeing of the animals (*Hobson & Stewart, 1997*). Bacteria are predominant in the rumen microbiome and are responsible for the conversion of indigestible plant biomass to energy and also aid in the formation of microbial protein; both processes drive the production efficiency of ruminants (*Firkins, 2010*). Interactions between different microbial domains

in the rumen play a significant role in determining the ruminal microbial ecology and their functional contribution to host metabolism (*Kumar et al., 2015*). Rumen microbial dynamics are influenced by a number of factors including host specificity, diet, age and the environment (*Edwards et al., 2004*). Elucidation of the interactions among microbial domains particularly in dairy cows has the potential to improve production such as feed efficiency and milk fat synthesis (*Weimer, 2015*). The transition period in dairy cows refers to a critical phase in the lactation cycle, lasting from three weeks before calving to three weeks post-calving, where the dairy cow experiences stress due to changes in diet, metabolism and physiological status. Although the dynamics of rumen bacteria during the transition period has received attention in the recent past (*Lima et al., 2015*; *Pitta et al., 2014*; *Wang et al., 2012*), further studies are required to understand the rumen microbial dynamics during the different phases of lactation across a large group of dairy cows.

Cultivation-independent approaches have greatly enhanced our knowledge on microbial diversity and also enabled us to assess their functional contribution to the host metabolism (*Stahl et al., 1988*). Particularly, next-generation sequencing (NGS) technology has enabled the sequencing of human and microbial genomes in a relatively short period of time (*Caporaso et al., 2011*). The most widely used high-throughput sequencing platforms available in the market include Roche 454 pyrosequencing, Ion Torrent Personal Genome Machine (PGM), and Illumina HiSeq (*Liu et al., 2012*). Although these platforms were originally tailored for large-scale operations such as whole genome sequencing, their bench-top versions (454 Jr, PGM, and MiSeq, respectively) have evolved since 2011 and have been extensively applied to bacterial genome sequencing (*Loman et al., 2012*). Since 2011, both MiSeq and PGM platforms have undergone improvements, including longer read lengths, more reads per unit cost, faster turn-around time, and a reduction in error rates (*Salipante et al., 2014*). Although a general comparison between these platforms has been reported (*Lam et al., 2012*; *Quail et al., 2012*), studies comparing the efficacy of these platforms on the same samples are limited (*Salipante et al., 2014*; *Scott & Ely, 2015*). As the Roche 454 platform phases out (*Fordyce et al., 2015*), there is a need for comparative studies that can aid in the transition from Roche 454 to other platforms.

The use of next generation platforms has greatly enhanced our knowledge of rumen microbes, their genes and enzymes (*Brulc et al., 2009*; *Hess et al., 2011*; *Jami, White & Mizrahi, 2014*). To date, there are nearly 55 research articles (based on Pubmed, 28 Jan, 2015) related to bacterial diversity from the rumen environment using Roche 454 while only 2 from MiSeq and 3 were reported based on Ion Torrent platforms. In an attempt to find a suitable alternative to the Roche 454 platform in relation to our microbial genomic work, we evaluated the use of Ion Torrent as an alternative to the 454 platform for the study of rumen microbial composition via 16S tag sequencing.

## MATERIALS AND METHODS

### Sample collection

Dairy cows that were donors of rumen fluid were maintained at Marshak farm and were maintained according to the ethics committee and IACUC standards for the University

of Pennsylvania (approval #804302). Four primiparous and four multiparous cows were sampled at three weeks prior to the anticipated calving date (S1), and again at 1–3 days post-calving (S2). Details of the animal experiment design, sampling protocol, and type of diet are described in a previous study (*Pitta et al., 2014*). Two samples were removed from the analysis due to a low number of reads in the Roche 454 sequencing run: one each from the primiparous and multiparous group in the pre-calving period. Thus, we analyzed a total of 14 samples, with six samples from pre-calving period and eight samples from post-calving period (Table S1).

## DNA extraction, PCR amplification, and 16S rRNA sequencing

The genomic DNA was extracted from all the rumen samples employing PSP Spin Stool DNA *Plus* Kit (Invitek, Berlin, Germany) using the protocol of *Dollive et al. (2012)*. The genomic DNA was amplified using the specific primers (27F) and BSR357, targeting the V1–V2 region of the 16S rRNA bacterial gene. The primer sequences and PCR conditions for Roche 454 are described in *Pitta et al. (2014)*. Though the primer sequences for Ion Torrent were similar to Roche 454, the forward primer carried the Ion Torrent trP1 (5′-CCTCTCTATGGGCAGTCGGTGAT-3′) and the reverse primer carried the A adapter (5′-CCATCTCATCCCTGCGTGTCTCCGACTCAG-3′), followed by a 10–12 nucleotide sample-specific barcode sequence and a GAT barcode adapter. The PCR mix was prepared using the Platinum PCR SuperMix High Fidelity kit (Invitrogen, Carlsbad, CA, USA). PCR conditions were the same for Roche 454 and Ion Torrent, as given by *Pitta et al. (2014)*. Amplicons of 16S rDNA were purified using 1:1 volume of Agentcourt AmPure XP beads (Beckman-Coulter, Brea, CA, USA). The purified PCR products from the rumen samples were pooled in equal concentration prior to sequencing in Roche 454 (Roche 454 Life Sciences, Branford, CT, USA) and Ion Torrent platforms.

## Bioinformatics and statistical analysis

To evaluate the similarities and dissimilarities between Roche 454 and Ion Torrent, we analyzed the 16S pyrosequence reads using the QIIME pipeline (version 1.8.0) (*Caporaso et al., 2010a*) and a small number of custom python scripts, followed by statistical analysis in R (*R Core Team, 2013*). Reads from both platforms were discarded if they did not match the expected sample-specific barcode and 16S primer sequences (forward and reverse primers), or if they were shorter than 50 bp or longer than 480 bp, or if they contained one or more ambiguous base calls. Reads were also discarded if a long homopolymer sequence was present; the threshold used was 5 bp for both platforms. Operational taxonomic units (OTUs) were formed at 97% similarity using UCLUST (*Edgar, 2010*). Taxonomic assignments within the GreenGenes taxonomy (12/10 release, *McDonald et al., 2012*) were generated using the RDP Classifier version 2.2 (*Wang et al., 2007*). We randomly sub-sampled (rarified) the resulting OTUs to 1,212 sequence per sample for both Roche 454 and Ion Torrent. Representative sequences for each were aligned to 16S reference sequences with PyNAST (*Caporaso et al., 2010b*). The resultant multiple sequence alignment was used to infer a phylogenetic tree with FastTree Price et al., 2010. To find shared OTUs between Roche 454 and Ion Torrent, we used a "closed reference" OTU

picking approach. To accomplish this, we identified Roche 454 OTUs that were shared between platforms, representative sequences from Roche 454 were compared to a reference database consisting of the Ion Torrent representative sequences. The same approach was adopted to identify shared OTUs in the Ion Torrent data. In this step, a sequence was considered a "hit" if it matched a sequence in the reference database at greater than 97% sequence identity.

Two measures of alpha diversity were calculated: Shannon entropy, an indicator of evenness in community structure, and richness, the number of OTUs observed. Analyses of community similarity ($\beta$-diversity) were performed for both the platforms separately by calculating pairwise distances using the phylogenetic metric UniFrac (*Lozupone & Knight, 2005*) on platform specific OTUs. We ran a Procrustes analysis of weighted UniFrac distance, comparing the principal coordinate matrices from Roche 454 and Ion Torrent. The goodness of fit ($M^2$ value) was measured by summing over the residuals, and significance was assessed by the Monte Carlo label permutation method (*Gower, 1975*).

Representative sequences from each OTU were chosen and taxonomy was assigned using the default methods in QIIME. To test for differences in taxon abundance, we normalized the abundances to the total number of reads in each sample (relative abundance). We considered the phyla appearing in at least 75% of samples. A generalized linear mixed-effects model was constructed with the lme4 package for R (*Bates et al., 2013*). The model used a binomial link function and included a random effect term for each animal. Study day was modeled as a continuous longitudinal variable with the following values: S1= $-3$ weeks, S2= 0.285 weeks.

## RESULTS

### Analysis of 16S sequence clusters

A total of 39,592 and 280,284 raw sequences were obtained from Roche 454 and Ion Torrent respectively, across a total of 14 samples on each platform. To minimize differences between the two platforms, we employed similar quality control protocols, including quality filtering, primer detection, and read demultiplexing. The number of reads recovered after quality filtering the reads of the 14 samples was 29,057 (73.39%) from Roche 454 and 203,910 (72.75%) from Ion Torrent (Table S1). Clustering with ULCUST generated 6,471 and 30,322 OTUs in Roche 454 and Ion Torrent, respectively. We randomly subsampled (rarified) the OTUs to 1,212 sequences per sample. Following the random subsampling step, we identified 2,824 shared OTUs from a total of 7,142 (40%) in the Roche 454 data. In the Ion Torrent data, we identified 3,118 shared OTUs from 9,813 OTUs (32%).

### Taxonomic comparisons

Taxonomic assignment of the OTUs identified 15 (Roche 454) and 18 (Ion Torrent) phyla in the rumen of cows used in this study. The three bacterial phyla that were recovered by Ion Torrent platform alone were Acidobacteria, GN02, and Verrucomicrobia (Table S2), which accounted for a very low abundance (0.01%) and were detected only in a few samples. The most abundant phyla in both the platforms were Bacteroidetes followed by Firmicutes, which together constituted over 90% of each sample in Roche 454 and over 86% in Ion
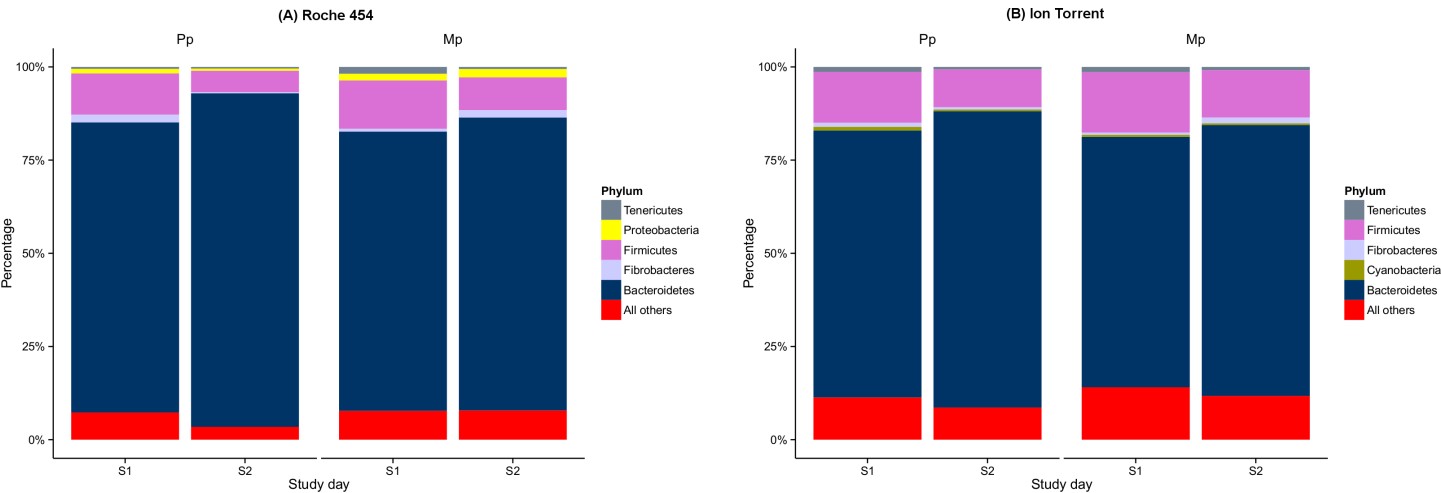

**Figure 1** Bacteria phyla during pre-calving (S1) and post-calving (S2) in Primiparous (Pp) and multiparous (Mp) dairy cows, as retrieved by the Roche 454 (A) and Ion Torrent (B) platforms.

Torrent (Fig. 1 and Table S2) The influence of study day (before and after calving) and study group (primiparous and multiparous cows) on the abundance of Bacteroidetes and Firmicutes appears to be similar for both Roche 454 and Ion Torrent data sets (Table 1). However, the influences of study group and study day in lower abundant phyla were different, for example the influence of study group and study day on Fibrobacteres is significant in Roche 454 whereas it is insignificant in Ion Torrent datasets.

We further compared the Roche 454 and Ion Torrent datasets at the genus level for the two major phyla observed (Bacteroidetes and Firmicutes). Genera with a proportion exceeding 1% in at least one sample were included in the analysis. In Roche 454, three genera from the Bacteroidetes group and one genus among Firmicutes showed differences by study group, whereas three genera from Bacteroidetes and four genera from Firmicutes showed differences by study day. In contrast, a similar analysis of Ion Torrent data revealed no differences within Bacteroidetes group and revealed differences among three genera in the Firmicutes with study group. With regard to study day, two genera in Bacteroidetes group and one genus in Firmicutes showed differences (Table 2). Notably, we found that BF311 (an uncultured genus of Bacteroidetes), *Mogibacteriaceae* (Firmicutes), and *Selenomonas* (Firmicutes) were detected only in the Ion Torrent platform.

Among Bacteroidetes members, the single most abundant genus in both the platforms was *Prevotella* (Roche 454: 81% and Ion Torrent: 75%; Fig. 4 and Table S2). The influence of study day on microbial composition of *Prevotella* was similar for both Roche 454 and Ion Torrent data sets, whereas, the influence of study group was observed only in Roche 454. In both platforms, most of the remaining sequences in the phylum Bacteroidetes were assigned either at the family or order level to unidentified unnamed species of the *Prevotellaceae* family or *Bacteroidales* order (Figs. S1 and S2). Genera in the Firmicutes phylum were also similar between the two platforms. The OTUs assigned to Firmicutes represented

Table 1 **Mean sample proportion of bacterial phyla by study group (Pp, primiparous and Mp, multiparous) and study day (S1, 3 weeks prior to calving; S2, 1–3 days after calving) in Roche 454 and Ion Torrent samples.**

| Phylum | Platform | Study group (SG) | | Study day (SD) | | P-value | | |
|---|---|---|---|---|---|---|---|---|
| | | Pp | Mp | S1 | S2 | $P_{SG}$ | $P_{SD}$ | $P_{SG*SD}$ |
| Bacteroidetes | Roche 454 | 0.8450 | 0.7700 | 0.7640 | 0.8400 | ** | *** | *** |
| | Ion Torrent | 0.7613 | 0.7032 | 0.6940 | 0.7610 | * | *** | *** |
| Cyanobacteria | Roche 454 | 0.0040 | 0.0030 | 0.0060 | 0.0020 | | *** | ** |
| | Ion Torrent | 0.0069 | 0.0050 | 0.0077 | 0.0047 | | | |
| Fibrobacteres | Roche 454 | 0.0110 | 0.0150 | 0.0140 | 0.0110 | ** | *** | *** |
| | Ion Torrent | 0.0082 | 0.0115 | 0.0086 | 0.0108 | | | *** |
| Firmicutes | Roche 454 | 0.0800 | 0.1060 | 0.1200 | 0.0730 | | *** | *** |
| | Ion Torrent | 0.1164 | 0.1420 | 0.1492 | 0.1142 | | *** | |
| Proteobacteria | Roche 454 | 0.0080 | 0.0200 | 0.0150 | 0.0140 | *** | ** | ** |
| | Ion Torrent | 0.0096 | 0.0097 | 0.0105 | 0.0090 | | | |
| Spirochaetes | Roche 454 | 0.0010 | 0.0040 | 0.0030 | 0.0020 | ** | | |
| | Ion Torrent | 0.0035 | 0.0073 | 0.0067 | 0.0044 | | | |
| Tenericutes | Roche 454 | 0.0050 | 0.0110 | 0.0120 | 0.0050 | | | * |
| | Ion Torrent | 0.0094 | 0.0109 | 0.0136 | 0.0076 | | ** | |
| TM7 | Roche 454 | 0.0030 | 0.0050 | 0.0050 | 0.0040 | ** | * | * |
| | Ion Torrent | 0.0050 | 0.0089 | 0.0093 | 0.0052 | * | ** | |

**Notes.**
The magnitude of the $p$-values for the effect of study group ($P_{SG}$), study day ($P_{SD}$), and the interaction term ($P_{SG*SD}$) are shown on the right.
[***] $P < 0.001$.
[**] $P < 0.01$.
[*] $P < 0.05$.

a substantial number of genera from *Clostridiales*, along with the *Lachnospiraceae, Ruminococcaceae* and *Veillonellaceae* families (Table 2; Figs. S3 and S4).

## Comparison between Roche 454 and Ion Torrent platforms
### Alpha diversity
The number of observed species per sample was higher for the Ion Torrent method, compared to Roche 454 (Fig. 2). The Shannon index value was similar between the two platforms at various sequencing depths, indicating a similar number of highly abundant species in both platforms.

### Beta diversity
We then quantified the resemblance between bacterial communities measured by Roche 454 and Ion Torrent. Weighted UniFrac distances for both communities were plotted using principal coordinate analysis (PCoA), and then the two PCoA plots were aligned using generalized Procrustes analysis (*Gower, 1975*). The aligned PCoA plot is visualized in Fig. 3. A comparison of pairwise sample distances by Procrustes analysis revealed moderate but statistically significant agreement in the microbial community composition between the two platforms ($M^2 = 0.36$; $P = 0.001$).

**Table 2** Mean sample proportion of bacterial genera by study group (Pp, primiparous and Mp, pluriparous) and study day (S1, 3 weeks prior to calving; S2, 1–3 days after calving) in 454 and Ion torrent samples.

| Taxon | Platform | Study group (SG) | | Study day (SD) | | P-value | | |
|---|---|---|---|---|---|---|---|---|
| | | Pp | Mp | S1 | S2 | $P_{SG}$ | $P_{SD}$ | $P_{SG*SD}$ |
| ***Bacteroidetes*** | | | | | | | | |
| Paraprevotellaceae | Roche 454 | 0.0040 | 0.0080 | 0.0040 | 0.0070 | | | |
| | Ion Torrent | 0.0070 | 0.0070 | 0.0070 | 0.0070 | | | |
| Bacteroidales | Roche 454 | 0.0420 | 0.0720 | 0.0620 | 0.0540 | ** | *** | *** |
| | Ion Torrent | 0.0430 | 0.0630 | 0.0560 | 0.0500 | | | |
| BF311 | Roche 454 | ND | ND | ND | ND | | | |
| | Ion Torrent | 0.0020 | 0.0040 | 0.0040 | 0.0020 | | | |
| CF231 | Roche 454 | 0.0130 | 0.0180 | 0.0150 | 0.0150 | | | |
| | Ion Torrent | 0.0160 | 0.0170 | 0.0150 | 0.0170 | | | |
| Prevotella | Roche 454 | 0.7200 | 0.5680 | 0.5820 | 0.6900 | ** | *** | *** |
| | Ion Torrent | 0.6280 | 0.5270 | 0.5250 | 0.6170 | | *** | *** |
| Prevotellaceae | Roche 454 | 0.0040 | 0.0060 | 0.0030 | 0.0070 | * | | * |
| | Ion Torrent | 0.0040 | 0.0050 | 0.0040 | 0.0040 | | | |
| RF16 | Roche 454 | 0.0350 | 0.0430 | 0.0610 | 0.0230 | | *** | *** |
| | Ion Torrent | 0.0360 | 0.0420 | 0.0520 | 0.0290 | | *** | |
| S24-7 | Roche 454 | 0.0150 | 0.0360 | 0.0140 | 0.0340 | | | * |
| | Ion Torrent | 0.0120 | 0.0190 | 0.0090 | 0.0200 | | | |
| YRC22 | Roche 454 | 0.0100 | 0.0100 | 0.0130 | 0.0080 | | | |
| | Ion Torrent | 0.0110 | 0.0110 | 0.0110 | 0.0110 | | | |
| ***Firmicutes*** | | | | | | | | |
| Mogibacteriaceae | Roche 454 | ND | ND | ND | ND | | | |
| | Ion Torrent | 0.0020 | 0.0030 | 0.0030 | 0.0020 | | | |
| Butyrivibrio | Roche 454 | 0.0040 | 0.0020 | 0.0030 | 0.0030 | | | |
| | Ion Torrent | 0.0070 | 0.0050 | 0.0060 | 0.0050 | | | |
| Clostridiales | Roche 454 | 0.0240 | 0.0290 | 0.0370 | 0.0180 | | *** | *** |
| | Ion Torrent | 0.0330 | 0.0380 | 0.0460 | 0.0270 | * | *** | * |
| Coprococcus | Roche 454 | 0.0020 | 0.0020 | 0.0030 | 0.0020 | | | |
| | Ion Torrent | 0.0060 | 0.0030 | 0.0040 | 0.0050 | | | |
| Lachnospiraceae | Roche 454 | 0.0130 | 0.0140 | 0.0150 | 0.0120 | | | |
| | Ion Torrent | 0.0210 | 0.0190 | 0.0230 | 0.0180 | | | |
| RFN20 | Roche 454 | 0.0040 | 0.0150 | 0.0130 | 0.0060 | ** | ** | |
| | Ion Torrent | 0.0040 | 0.0150 | 0.0110 | 0.0090 | * | | |
| Ruminococcaceae | Roche 454 | 0.0060 | 0.0120 | 0.0140 | 0.0060 | | ** | |
| | Ion Torrent | 0.0080 | 0.0210 | 0.0170 | 0.0120 | *** | | |
| Ruminococcus | Roche 454 | 0.0040 | 0.0080 | 0.0070 | 0.0050 | | | |
| | Ion Torrent | 0.0100 | 0.0110 | 0.0110 | 0.0100 | | | |
| Selenomonas | Roche 454 | ND | ND | ND | ND | | | |
| | Ion Torrent | 0.0030 | 0.0020 | 0.0010 | 0.0040 | | | |

**Table 2** (*continued*)

| Taxon | Platform | Study group (SG) | | Study day (SD) | | P-value | | |
|---|---|---|---|---|---|---|---|---|
| | | Pp | Mp | S1 | S2 | $P_{SG}$ | $P_{SD}$ | $P_{SG*SD}$ |
| Succiniclasticum | Roche 454 | 0.0110 | 0.0070 | 0.0130 | 0.0060 | *** | | |
| | Ion Torrent | 0.0100 | 0.0070 | 0.0110 | 0.0070 | | | |
| Veillonellaceae | Roche 454 | 0.0020 | 0.0020 | 0.0010 | 0.0030 | | | |
| | Ion Torrent | 0.0030 | 0.0040 | 0.0030 | 0.0040 | | | |

**Notes.**

ND, Not Detected; The magnitude of the *p*-values for the effect of study group ($P_{SG}$), study day ($P_{SD}$), and the interaction term ($P_{SG*SD}$) are shown on the right.

*** $P < 0.001$.
** $P < 0.01$.
* $P < 0.05$.

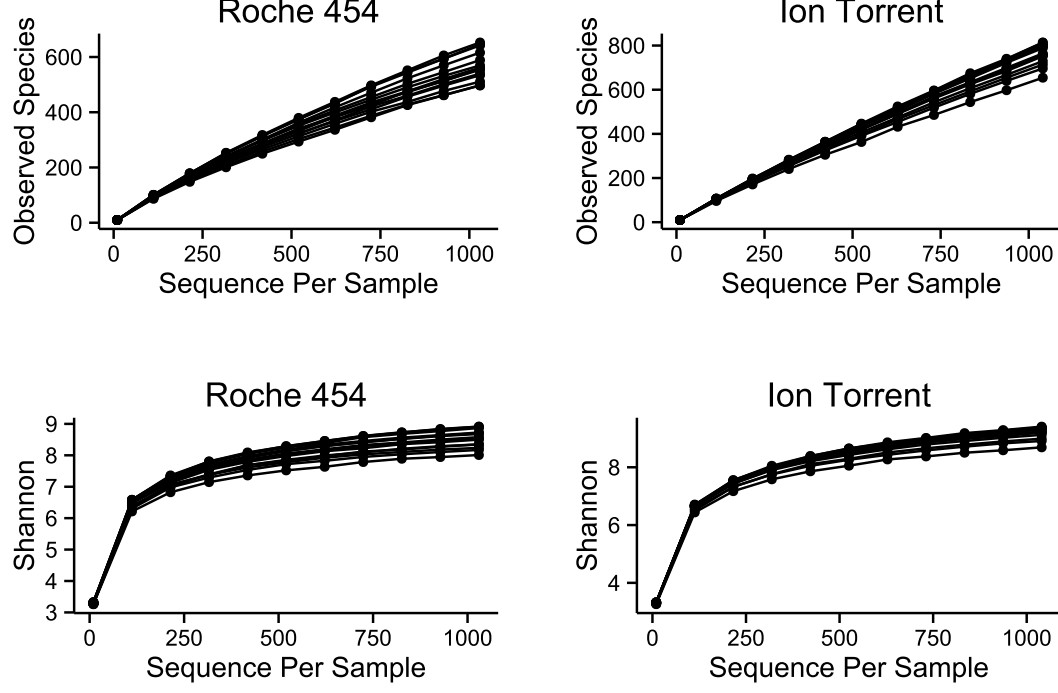

**Figure 2   Alpha diversity of samples sequenced on the Roche 454 and Ion Torrent platforms.** The rarefaction curves show the mean value for number of OTUs observed and Shanonn diversity at various sequencing depths.

## Taxonomic comparison at genus level

A sample-to-sample to comparison at the genus level between the two platforms was performed using chi square test (Table S3). For this analysis, twenty nine genera were compared by sample for both platforms (Table S3). Of these, only *Prevotella* from Bacteroidetes phylum, and one unclassified bacterial lineage were found to be different between the two platforms in more that 50% of samples ($p < 0.05$; chi square test). A majority of the remaining genera were found to be different between only one or two samples. Thus, the composition of bacterial communities was generally consistent between platforms, with two notable exceptions.

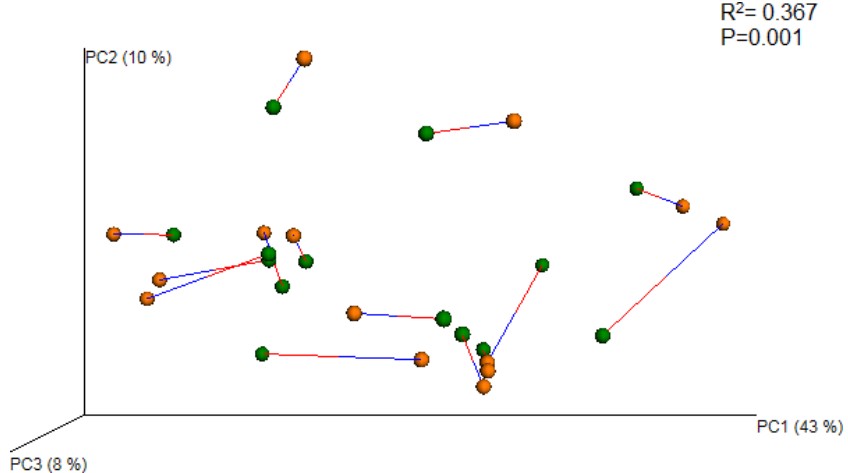

**Figure 3 Procrustes analysis of Roche 454 and Ion Torrent samples.** This analysis compares the principal coordinates analysis (PCoA) of UniFrac distance between samples for each platform, showing the best superimposition of one plot on the other. Samples in Roche 454 (orange circle) and Ion Torrent (green circle) are connected by a line. The blue line indicates Roche 454 samples and the red line indicates Ion Torrent samples. A lower distance between the circles indicates a higher degree of concordance between the plots.

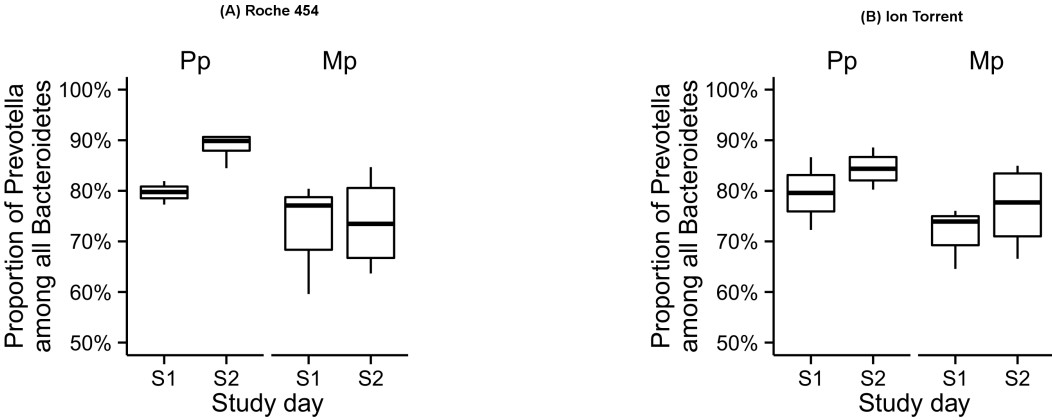

**Figure 4 Proportion of Prevotella among all Bacteroidetes in the rumen samples collected during the pre-calving (S1) and post-calving (S2) in both Primiparous (Pp) and multiparous (Mp) dairy cows, as retrieved by the Roche 454 (A) and Ion Torrent (B) platforms.**

## DISCUSSION

The introduction of NGS technology has had a dramatic effect on researchers' ability to study bacterial communities via DNA sequencing. The throughput has increased by 500,000-fold and the number of reads per genome is increasing 100-fold every year, yet the cost of sequencing is reducing by half every 5 months (*Baker, 2010*). The rumen microbiome of herbivores is a classic example that has been explored in detail using different NGS approaches (*De la Fuente Oliver et al., 2014*; *Jami, White & Mizrahi, 2014*; *Peng et al.,*

*2015*; *Pitta et al., 2014*; *Pitta et al., 2010*), but several inconsistencies among these reports prevail, some of which may be attributed to differences in the approach employed. In this study, we attempted to account for biases that could be introduced due to different NGS platforms while making comparisons between 16S rDNA bacterial profiles in the rumen of dairy cows. We concluded that, although the percentage of common sequences was low between the platforms, the microbial fingerprints and phylogenetic composition retrieved by both Roche 454 and Ion Torrent platforms are comparable, with minor exceptions.

### Ion Torrent vs. Roche 454 Roche platforms

Considering the limitations such as low quality reads, variations in read length, and higher number of homopolymer sequences associated with Ion Torrent (*Loman et al., 2012*), we leveraged to extract more number of sequences per sample in Ion Torrent platform with the result that throughput was several fold higher in magnitude as compared to Roche 454 datasets. High quality reads with greater read lengths despite lower throughout from Roche 454 platform is the most preferred and desirable feature for microbial diversity studies (*Brulc et al., 2009*). Both the platforms delivered long reads, but the read length of Ion Torrent had a wider range compared to Roche 454 (Roche 454; 53–492, Ion Torrent; 8–559). In our study, we observed that a greater proportion of sequences were eliminated at 3 and 4 homopolymer length while a greater proportion of sequences were retained at 5 homopolymer length. For downstream analysis, we recovered 29,057 (73.39%) from a total of 39,592 raw reads in Roche 454 platform and 203,910 (72.75%) from a total of 280,284 raw reads in Ion Torrent platform at 5 homopolymer length. To account for unequal distribution of reads per sample between platforms, (1,212–3,036 in Roche 454 vs. 5,653–27,336 in Ion Torrent), we adopted normalization through subsampling of reads at the minimum sequencing depth for both platforms. This additional step was performed to avoid the influence of this differential distribution of reads on the bacterial composition.

### Ruminal Bacterial diversity dynamics

The effect of diet and age on the community composition, evident in our previous study using Roche 454 platforms (*Pitta et al., 2014*), was reproduced when the same set of DNA samples were sequenced on Ion Torrent. This increases confidence that the two platforms will yield similar results in future studies of bacterial communities in the rumen. The finding is reinforced by Procrustes analysis, which showed that overall community composition was similar between both platforms. Although the overall level of similarity was not great enough to combine data from two platforms in a single analysis of UniFrac distance, we determined that independent distance analyses were reproducible between platforms. Further, we were able to infer that increasing the sequence depth did not unduly influence community profiles when analyzed by UniFrac distance, reinforcing the findings of *Caporaso et al. (2011)*.

The taxonomic composition of bacterial communities at the phylum level recovered by Ion Torrent is congruent with the Roche 454 data. The most abundant phyla were Bacteroidetes followed by Firmicutes, which together constituted over 90% of each sample in Roche 454 and over 86% in Ion Torrent. The phylum-level abundance observed

here agrees with other reports that have employed Roche 454 (*Pitta et al., 2014*; *Pitta et al., 2010*) and also Ion Torrent (*De la Fuente Oliver et al., 2014*). In addition to these phyla, Fibrobacteres, Proteobacteria and Tenericutes, Actinobacteria and Spirochaetes are commonly reported in rumen samples from dairy cows, but contribute to a very low abundance. This observation was also recapitulated in our findings for both platforms. Additionally, the 16S rDNA-based taxonomic composition of bacterial communities retrieved here is in accordance with other shotgun metagenomic datasets sequenced on the Ion Torrent platform (*Patel et al., 2014*; *Pitta et al., 2015*; *Singh et al., 2014*).

Differences between datasets from the two platforms were evident in phyla that contributed to less than 5% of the populations. A sample-to-sample comparison revealed differences in phylogenetic composition from family and beyond. For example, at the genus level, although *Prevotella* was abundant in both platforms, the lower abundance of *Prevotella* observed in Ion Torrent compared to Roche 454 may be due to an increase in number (14) of several genera that were not detected in the Roche 454 data. Despite these differences at the genus level, the effect of age and diet on different bacterial genera in the rumen of dairy cows was consistent between the two platforms.

## CONCLUSIONS

It has become evident in the recent past that the rumen microbiome plays a significant role in improving the production efficiencies of dairy cows. As next-generation sequencing platforms and chemistries continue to expand and improve, we expect that major advancements in sequencing may contribute to significant improvements in dairy production through improving nutrition and management that support a production efficient microbiome. While different sequencing platforms are applied to explore the rumen microbiome, it is imperative that studies should also compare and contrast findings from different platforms to avoid discrepancies across rumen microbiome studies. The results presented here show that major conclusions based on Roche 454 data are also reproduced on the Ion Torrent platform. Thus, we found that the Ion Torrent platform is a suitable option for future rumen microbiology studies, provided that researchers are consistent in DNA extraction methods, PCR protocols, and bioinformatics pipelines.

## ACKNOWLEDGEMENTS

We are thankful to the Biomedical Research Core Facilities, University of Pennsylvania, for sequencing services.

### Funding

Part of the funding for this study came from internal grants from Center for Host-Microbe interactions, University Research Foundation from the University of Pennsylvania. The funders had no role in study design, data collection and analysis, decision to publish, or preparation of the manuscript.

## Grant Disclosures

The following grant information was disclosed by the authors:
Center for Host-Microbe interactions, University Research Foundation.

## Competing Interests

The authors declare there are no competing interests.

## Author Contributions

- Nagaraju Indugu analyzed the data, contributed reagents/materials/analysis tools, wrote the paper, prepared figures and/or tables, reviewed drafts of the paper.
- Kyle Bittinger analyzed the data, wrote the paper, prepared figures and/or tables, reviewed drafts of the paper.
- Sanjay Kumar wrote the paper.
- Bonnie Vecchiarelli performed the experiments, contributed reagents/materials/analysis tools, wrote the paper.
- Dipti Pitta conceived and designed the experiments, wrote the paper, reviewed drafts of the paper.

## Animal Ethics

The following information was supplied relating to ethical approvals (i.e., approving body and any reference numbers):

Dairy cows that were donors of rumen fluid were maintained at Marshak farm and were maintained according to the ethics committee and IACUC standards for the University of Pennsylvania (approval #804302).

## Data Availability

The raw sequence data and corresponding mapping files are available in figshare.com (http://figshare.com/articles/comparison_of_454_and_PGM_platforms/1536771).

## Supplemental Information

Supplemental information for this article can be found online at http://dx.doi.org/10.7717/peerj.1599#supplemental-information.

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
