# Peer review of "A comparison of rumen microbial profiles in dairy cows as retrieved by 454 Roche and Ion Torrent (PGM) sequencing platforms"

_PeerJ, doi:10.7717/peerj.1599_

## Round 0.1 · original submission · Minor Revisions

Please revise the manuscript as suggested by the reviewers.

·

Basic reporting

Overall, the manuscript is well-written and authors have tried to cover all the points. However, there are several spelling errors in the manuscript. Apart from that, various points needs to be addressed.
1. Consistency in the manuscript. Consistency in naming the sequencing platform was lacking throughout the manuscript. Throughout the manuscript, authors have mentioned Roche 454 GS Jr as Roche or 454 or 454 pyrosequencing. However, they have mentioned the name of system i.e., PGM at all places in case of Ion Torrent. So, both the platforms should have common nomenclature patterns throughout the manuscript. Roche 454 and Ion Torrent are platforms while GS Jr and PGM are their respective systems.
2. In Figure 3, same phyla (and their colours) should be considered for comparing two platforms.
3. In Supplementary Figure S1, change the scale to log10 for better viewing. In your image, 3 phyla detected only on Ion Torrent are also visible on 454 bars.
4. Label of Supplementary Figure S3 and S4 are exchanged.
5. In Supplementary table S1, mention actual abundance in both the platforms.
6. In table 1 and 2, what is difference between value 0 and ND? Give description of * in terms of different level of significance in caption.

Experimental design

Overall, experimental design considered for comparing two platforms is appropriate. However, points are lacking in the manuscript:
1. Mention clearly about the number of animals used per group (primiparous and multiparous).
2. The criteria for filtering out the reads is not proper. Maximum length to be filtered is kept at 50-100 bp longer than expected product size. However, authors have kept it at 1000 bp. Also, similar constraints needs to be used for homopolymer stretch, 6 bases, for 454 and Ion Torrent data.
3. The criteria for “common” reads is not justifiable. Even while matching reads against barcode sequences, threshold is kept lesser than 100%. Also, because of the differences in the read lengths of both platforms, chances of obtaining exactly identical sequences will be very low. This is also reflected in terms of the actual percent of common reads in both the platforms (1% in 454 & 0.5% in Ion Torrent) and also obtaining similar profiles of bacteria in both the platforms. Hence, common reads concept should be relaxed (in terms of composition and length) or omitted.
4. Include the information regarding the number of sequencing runs carried out as well as total data obtained with some details about average read length, modal read length and quality. Also, provide the rarefaction curves and α-diversity for samples.
5. Mention number of “common” OTUs and platform-specific OTUs. Word “shared” OTUs will be more appropriate than “common” OTUs.

Validity of the findings

There are several concerns regarding the flow of analysis used in the manuscript. The data obtained from both the platforms is highly uneven (around 5x more in case of Ion Torrent than 454), still data was not normalised in any manner for comparison. At many places, authors have compared number of reads between both the platforms without normalising the number. There are some discrepancy while describing the results of Figure 1 & 2 in results and discussion section.
1. Results regarding filtering among both the platforms should be shown in depth i.e., number of reads filtered because of each parameter.
2. Non-uniform distribution of reads in Ion torrent can be caused due to handling errors also. However, data needs to be rarefied before using for any analysis.
3. Percentage of common reads may differ because of the uneven number of reads taken for analysis.
4. In Figure 1 and Figure 2, number of reads are plotted on Y-axis and compared between both the platforms. However, comparison should only be made between percentage of reads or only if data is normalised which is not the case. Also, the representation of bar graph is not appropriate.

·

Basic reporting

The manuscript is well written, with proper introduction to the background and proposition of the need for current work.

Experimental design

No comments

Validity of the findings

Please see my comments in the attachment to clarify some of the statements in Methods and Results section (also pasted in here:).

In the manuscript, Indugu et al, describe their work on the comparison of 454 Roche and Ion Torrent (PGM) sequencing platforms using rumen microbial profiles. This work is important in the context of current scenario, where popular old technologies are being phased out and emergence of newer technologies poses questions of their reliability and comparability to the established standards. As the usage of 454 Roche is under decline and the company already announced shutdown in mid--‐2016, comparative studies like the current one gains importance.

The authors tried to reduce the influence of platform specific variables so as to make a meaningful comparison. However, there were some inconsistencies that need to be addressed.

Lines 89 to 95

Though the authors reference a previous publication for the experimental design, it was not clear from the text as well as the reference how many samples (totaling 14) were used per each group of pP, mP, S1 and S2.

Line 122 to 128 and Line 146 to 157:

The sequence of operations from raw reads until OTU generation is not clearly explained and inconsistent between the Methods section (line 122 to line 128) and Results section (line 146 to line 157).

In methods section the authors state that reads from 454 and PGM were first combined together, de--‐replicated using QIIME, and OTUs generated using UCLUST. Subsequently OTUs were divided into platform specific and common groups.

In the results section however, they mention clustering by UCLUST to first generate OTUs. Subsequently they mention that reads are combined from 454 and PGM and clustered by QIIME (line 151).

Though it was not clearly mentioned, it seems that the 6772 and 22337 OTUs (line 150) for 454 and PGM respectively, denote the platform specific OTUs. Right? Lines 158 to 160 then describe how the common and platform specific OTUs were determined. It will be informative to state number of common OTUs.

How were the percentages of common reads (line 157) obtained? Are these the percentage of filtered (or raw) reads in each platform or the combined reads? Also it is not clear (from differing sentences in methods and results sections) whether the common reads were identified before OTU identification or after.

The authors should attempt to clarify and reframe the sentences in this portion of the text.

Line 158 to 162:

In figure1 representing the distribution of number of reads in common OTUs, what does the authors mean by “OTU size” on x--‐axis? The authors should explain the scale used on the x--‐axis. If the x--‐scale just represents the OTUs in decreasing order of their size (number of reads), then a better label would be “OTU rank”. It would also be needed to start the x--‐scale from 1 rather than 0. Only the first 500 are represented in the figure, so it would be good to mention in the text (as stated in the previous comment) the total number of common OTUs.

Another question arises related to this figure. The authors stated that there were 3151 common reads in 454 and 8582 common reads in PGM. The figure however doesn’t seem to depict this large difference in number of reads ?

Line 163 to 168 (Figure 2)

Looking at Figure2, it is unclear how some platform specific OTUs are 97% to 100% identical to the common OTUs (Figure 2). Since the OTUs were calculated based on 97% similarity, in theory, sequences belonging to a different OTU should be less than 97% identical.

Also why did the authors state that the platform specific OTUs in PGM as less similar to common OTUs, while the figures shows an opposite case?

Lines 171 to 190

The results showed similar trends in abundances between the study groups, in 454 and PGM for the most abundant taxa. But for the low abundant ones there were differences observed. There was no mention of any data normalization in the methods. Would the results be better if any normalization performed on the data?

A good initial indicator to compare the overall species composition in the two platforms would be to do calculate alpha diversity using any of the several methods.

Lines 204 to 209

Procrustes analysis (in beta diversity) indicated a good agreement between the two platforms, were only the common OTUs used for this analysis, or the all the OTUs in each platform?

Lines 240 to 241

It would be good to indicate the number of raw reads obtained in each platform.

Lines 249 to 255

The authors mention about the common reads between the platform. Rather than common reads it would be sufficient to just talk about the common OTUs. As it is expected to not to get exactly same reads in sequencing. Comparison of OTUs however will be more meaningful (the authors did this already).

Lines 287 to 298

The authors conclude that PGM compares well with the results obtained in 454. Addressing the above comments and clarifying some of the inconsistencies would further strengthen these conclusions.

---

## Round 0.2 · accepted · Accept

Congratulations! Your manuscript is accepted

·

Basic reporting

No Comments

Experimental design

No Comments

Validity of the findings

No Comments